# Testing the Effectiveness of a Blended Intervention to Reduce Suicidal Ideation among School Adolescents in Chile: A Protocol for a Cluster Randomized Controlled Trial

**DOI:** 10.3390/ijerph19073947

**Published:** 2022-03-26

**Authors:** Daniel Núñez, Jorge Gaete, Daniela Meza, Javiera Andaur, Jo Robinson

**Affiliations:** 1Faculty of Psychology, Universidad de Talca, Talca 3460000, Chile; danielamezalorca6@gmail.com (D.M.); javiera.andaur.c@gmail.com (J.A.); 2Millennium Nucleus to Improve the Mental Health of Adolescents and Youths, Imhay, Santiago 8320000, Chile; 3Associative Research Program, Center of Cognitive Sciences, Faculty of Psychology, Universidad de Talca, Talca 3460000, Chile; 4Faculty of Education, Universidad de los Andes, Monseñor Álvaro del Portillo, 12455 Santiago, Chile; 5Orygen, Parkville, VIC 3052, Australia; jo.robinson@orygen.org.au; 6Centre for Youth Mental Health, University of Melbourne, Parkville, VIC 3010, Australia

**Keywords:** suicidal ideation, adolescents, cognitive-behavioral therapy, blended intervention, randomized controlled trial, protocol

## Abstract

Suicidal ideation is prevalent in adolescents and is a marker for subsequent psychiatric vulnerability and symptom severity. Literature shows that blended care (integrating online and offline components in a treatment process) could improve the effectiveness and adherence of interventions targeting suicidal ideation in adolescents, but the evidence is inconclusive. Thus, we will test the effectiveness of a blended intervention to reduce suicidal ideation (primary outcome) in school settings using a single-blind two-armed cluster randomized controlled trial (cRCT). The internet-based component corresponds to the Reframe-IT, a program encompassing eight online sessions based on cognitive-behavioral therapy (CBT) principles. The face-to-face intervention will be delivered through four CBT sessions. Additionally, we will assess the effect of the intervention on the following secondary outcomes: suicidal attempts, depressive symptoms, hopelessness, emotional regulation, and problem-solving skills. Primary and secondary outcomes will be assessed at post-intervention, 3-month, 6-month, and 12-month follow-up. Finally, we will explore the mediation role of cognitive, emotional, and behavioral correlates of suicide on the effect of the intervention. Results will inform whether the intervention can reduce suicide among school adolescents and be implemented on a large scale in Chile.

## 1. Introduction

### 1.1. Background and Rationale

Suicide-related behaviors (SRB) including suicidal ideation (SI) and suicide attempts (SA) are common in adolescents, representing a major health burden [1,2]. The lifetime and 12-month worldwide prevalence of suicidal ideation in this population is 18% and 14.2%, respectively [3]. It is well known that youths are reluctant to seek professional help [4] and that the capability for the early detection of individuals at risk for suicide is low [5]. Accordingly, the literature recommends using evidence-based strategies to actively detect at-risk subjects and provide interventions for reducing suicide risk. Most adolescents receive a formal education [6]. Moreover, school-based interventions can positively impact SRB and overall do not appear to cause harm [7]. Thus, schools are currently considered an accepted context to implement suicide-prevention programs [8,9]. Recently, strategies have called for delivering interventions in face-to-face and digital settings [10]. The evidence shows that combining these two approaches (blended care), could improve their effectiveness [11] and reduce health-related costs; however, further research is needed, mostly in young people from Latin American countries [7].

There has been a rapid growth in the development and usage of internet-based interventions to prevent and treat mental disorders [12]. Digital technologies are currently suitable for improving the appeal, cost, and reach of mental health interventions [13,14]. This is very important for low and middle-income countries [10,15], where the treatment gap of mental disorders is very high [16,17]. Moreover, it is also relevant for improving the treatment for underserved populations [18], such as adolescents [19], who cannot access timely and frequent mental health services [20,21]. Cumulative evidence shows that digital interventions might overcome structural and attitudinal barriers associated with accessing mental health care [22], and that might be effective for reducing SI [23] and repetition of self-harm [24]. Internet interventions based on cognitive-behavioral therapy (CBT) principles have been one of the most studied technology-assisted interventions for mental health, with a substantial part of the interventions targeting depression and anxiety disorders in adults [14]. Some research shows that, when carefully designed, these interventions could improve adherence, satisfaction, and clinical outcomes in adolescents with anxiety and depression [25,26,27,28,29]. However, further research is needed in low- and middle-income countries [30,31,32].

CBT-based interventions seem to be the most promising approach to reduce SRB [33], including SI in adolescents [34,35], but some challenges must be faced to effectively integrate evidence-based programs into real-world adolescent settings [36]. First, online interventions to reduce SRB among adolescents are scant and primarily conducted in developed countries [7]. Two recent reviews [23,37] identified four studies. Van Voorhees et al. [38] observed a reduction in depressive symptoms (DS), yet significant decreases in SI and hopelessness in both control and experimental groups. The protocol by Stallard et al. [39] was reported as a useful and acceptable strategy to manage self-harming thoughts. The community-based pilot RCT by Hill and Pettit [40] yielded reduced perceptions of burdensomeness but not in SI. Finally, Perry et al. [29] observed reduced DS by a cluster randomized controlled trial (cRCT) in educational settings. Although these studies have shown some positive results, only Stallard et al. [39] targeted SRB as the primary outcome.

Second, most of the school-based programs are universal and selective interventions [6]. Most internet-based programs are indirect interventions (targeting depression, anxiety, hopelessness, or burdensomeness). The review by Büscher et al. [41] showed that the only direct intervention targeting suicidal ideation in school students with suicidal ideation is the Reframe-IT, developed by Robinson et al. [42]. This intervention was built following the main principles of CBT, considering the evidence that cognitions are central in the development and maintenance of mental health difficulties and that both emotional and behavioral problems can be resolved by modifying dysfunctional thoughts [43]. The intervention is organized in eight modules to promote the following skills: problem identification, emotional recognition, distress tolerance, behavioral activation, cognitive restructuring, and problem-solving skills. Following international recommendations [44], secondary outcomes such as depression and hopelessness were additionally assessed [45]. The pilot study (pretest/post-test design with an 8-week intervention phase) developed in 11 secondary schools showed that the program was enjoyable, helpful, and acceptable by participants [46] and effective in reducing SI, DS, and hopelessness at post-intervention [47]. The RCT study by Hetrick et al. [48], found improvement in SI levels (intervention group: mean = −61.6, SD 41.6; control group: mean = −47.1, SD 42.3). These differences were not significant, which might be due to the small sample size [49]. Given preliminary results showing that it is a safe online intervention to engage with suicidal young people, further research testing its effectiveness with larger samples is guaranteed [7]. 

Third, prior research has found that direct face-to-face interventions effectively reduced SI in adults [50] and adolescents [51]. The underlying mechanisms explaining this preliminary effect remain uncertain. Very few studies have proposed specific processes [52] beyond anxiety, depression, and hopelessness [23]. The review by Milner et al. [53] found that social support, suicide prevention literacy, and learning alternative coping behaviors could be mechanisms associated with reduced SRB after Brief Contact Interventions (BCIs). This fits with Iyengar et al. [52], who observed that the processes related to reductions in SRB were individually self-driven (behavioral self-regulatory processes) and socially driven (family or social support networks) and that the central components of effective interventions are emotion regulation, problem-solving, and communication skills.

Fourth, digital interventions have shown similar (small and significant) effects to face-to-face programs [37], but the knowledge on processes influencing these effects is limited. Besides the abovementioned processes, human guidance has been suggested as a potential factor in improving internet-based psychological treatment effectiveness [54]. Guided internet-based interventions have been associated with good outcomes and treatment engagement in adolescents and young people, at least for depression and anxiety [31]. Cumulative evidence shows that adherence and effectiveness of these interventions could improve by increasing motivation and interest through strategies that allow users to tailor activities to their natural context and promote ongoing use [55]. Additionally, periodic monitoring of therapeutic activities [56] and reinforcing therapeutic alliance [33] have been reported as relevant factors. This is supported by recent research showing that treatment completion rates could rise when interventions are delivered with real-time guidance in contexts such as schools [26], and that the impact of interventions could be higher if accompanied by a therapist [13]. Thus, it is currently accepted that blended care (the integration of online and offline components in a treatment process) [57] represents a suitable manner to favor the involvement of therapists in internet-based programs, which has been suggested as a potential factor of success for internet-based interventions [14]. 

In summary, there is a recognized need to improve the effectiveness of interventions targeting suicide ideation in adolescents. Recent evidence shows that including face-to-face intervention could improve the effectiveness of these interventions. In this context, we will develop a protocol of a blended intervention combining an internet-based intervention (Reframe-IT) with four face-to-face sessions based on CBT principles. 

### 1.2. Objectives

The main objective of this study is to develop a blended intervention to reduce suicidal ideation and to test its effectiveness among adolescents in secondary schools in Chile at post-intervention and 3-month, 6-month, and 12-month follow-up. Additionally, the study will identify and compare changes in depressive and anxiety symptoms, hopelessness, emotional regulation, and problem-solving skills among adolescents in the intervention group schools and the control group. Finally, we will explore the influence of potential mediators on the observed effects of the integrated blended intervention. 

We hypothesize that the intervention would reduce suicidal ideation among the treatment group compared to the control group at post-intervention. Specifically, we hypothesize that the intervention will lead to the following changes among the treatment group compared to the control group at post-intervention and follow-up: reduced feelings of depression, anxiety, and hopelessness, increased emotional regulation, and problem-solving skills. Finally, we hypothesize that the effect of the integrated blended intervention on suicidal ideation will be mediated by hopelessness, emotional regulation, and problem-solving skills, and partially mediated by CBT skills acquired from the intervention. The results of this study will provide relevant insights to reinforce direct and indicated interventions in educational settings. 

### 1.3. Trial Design

This is a protocol for a single-blind, two-armed randomized controlled trial evaluating changes in primary and secondary outcomes post-intervention and follow-up. The two arms will be: (a) the intervention, which comprises eight internet-based modules of CBT delivered over 10-weeks using the program Reframe-IT and four 45-min face-to-face psychotherapeutic CBT sessions, and (b) control group (Treatment-As-Usual) (see Table 1).

## 2. Methods: Participants, Interventions, and Outcomes

Trial registry: ClinicalTrials.gov: NCT05229302. Registered 8 February 2022.

### 2.1. Study Setting

Participants will be adolescents attending schools with secondary education (grades 9–11), mixed sex, located in three different cities in the South of Chile, with at least two classes per year. We expect to recruit at least ten schools per arm. Each arm should include 140 eligible students.

#### Eligibility Criteria

Inclusion criteria for the present study are:Secondary students attending grades 9–11.High scores in suicidal ideation, last month (score ≥ 3 in the Columbia Suicide Severity Rating Scale (C-SSRS) [58].Fluent in the Spanish language.

The exclusion criteria are:Suicide attempt(s), last month.Severe depressive symptoms assessed by the PHQ-9 (>19 points).High severity of psychotic symptoms assessed by the Community Assessment of Psychic Experiences-Positive scale (CAPE, P15) [59]; (cut-off = 1.47) [60].

### 2.2. Recruitment and Informed Consent

The schools will learn about the study through invitations made to their institutional email and social networks. Eligible schools will be invited to participate through letters sent to the authorities, who will sign an agreement letter to participate in the study. This research project has the approval of the Scientific Ethics Committee of the Universidad de Talca (12 May 2021). At the beginning of the academic year, we will send a letter to the emails with information about the study, inclusion and exclusion criteria, screening instruments, the selection process, and the written consent to all parents or main caregivers of students. The informed and written consent will be signed digitally through Qualtrics. Qualtrics is a platform that allows information to be collected online with high security standards and that also can facilitate the digital signature through a cell phone or tablet (using the finger) or through a computer (with a mouse). Additionally, all students attending grades 9–11 of selected schools will be invited to participate through informative talks about the study. All students will be asked for written assent. All consented/assented students will be surveyed with the following screening questionnaires during the second month of the academic year: the Columbia Suicide Severity Rating Scale (C-SSRS) [58]; PHQ-9 [61]; and the Community Assessment of Psychic Experiences-Positive (CAPE-P15) [59]. The Qualtrics platform will also be used to collect data of screening instruments: suicidal ideation (SI) and suicide attempts (C-SSRS), depressive symptoms (PHQ-9), and psychotic symptoms (CAPE-P15). These data will be used for the study eligibility analysis. All who report any level of SI during the previous month (score ≥ 3 in the C-SSR) will be invited to participate in the following step of the study. Informed consent will cover general screening, baseline measurements, and follow-up during and after the intervention.

### 2.3. Additional Consent Provisions for Collection and Use of Participant Data and Biological Specimens

Not applicable. No biological samples will be collected.

## 3. Interventions

### 3.1. Explanation for the Choice of Comparators

All participants in the intervention group will be sent to primary care centers, where a protocol for managing suicidal ideation defined by the Chilean Ministry of Health will be applied [62]. The whole procedure will be described as Treatment-As-Usual (TAU). TAU consists of identification and assisted referral to a primary care clinic, where trained psychologists will assess the symptomatology and propose a course of actions, from initiation of psychotherapy to referral to a general practitioner to initiate medication if needed. Psychotherapy in primary care clinics consists of a range of 4–8 sessions twice a month. Most of the content of the psychotherapy is based on the Cognitive-Behavioral Model. When needed, general practitioners may suggest using medications, which generally are SSRI antidepressants (e.g., Fluoxetine, Sertraline). A medical check-up is conducted every month or every two months.

The control group will receive the Treatment-As-Usual (TAU). In order to reduce the risk of performance bias, all adolescents who need to have care in a public health center will receive the same treatment. The public health centers have similar structures and resources, they have standardized protocols for managing mental disorders and suicide risk and the professionals have similar backgrounds and skills. To test potential biases, we will register and quantify all different interventions delivered by health centers.

### 3.2. Intervention Description

#### 3.2.1. Experimental Intervention

##### Internet-Based Component (Reframe-IT)

It has been described elsewhere [46,48]; it comprises eight modules of CBT delivered over a 10-week intervention period. The program will be administered at schools by trained school psychologists (one psychologist per school). The authors will deliver the training of the program. Each participant will have access to their personalized web page accessed via secure login. Each module will be completed in the psychologist’s presence and participants will also be able to access it at home 24 h a day. For safety reasons, the program does not have social networking functions. Standard CBT approaches are delivered in the intervention, focusing on SI and behaviors. The modules and their respective topics are: -Module 1: engagement and problem identification;-Module 2: emotional recognition and distress tolerance;-Module 3: identification of automatic negative thinking;-Module 4: behavioral activation help seeking;-Module 5: behavioral activation activity scheduling (including relaxation techniques);-Module 6: problem solving;-Module 7: cognitive restructuring;-Module 8: wrap-up session.

These contents are delivered via a series of video diaries by young people as main characters with an adult ‘host’ character highlighting the CBT contents presented in the context of the video diaries. There are two activities per module. Overall, each module lasts about 25 min. The site has a message board through which the participant could communicate with research therapists (clinical psychologists), who also check completed activities and respond with personalized but standardized messages. Finally, a series of fact sheets cover various related topics including managing SI, plus downloadable relaxation MP3s.

##### Face-to-Face Intervention

We developed four face-to-face CBT sessions following a similar structure of the Reframe-IT sessions (brief videos, activities, and mood checking). Each session lasts about 45 min. The aims of the sessions are described as follows:

Session 1: To improve the participant’s literacy on emotions, reinforcing the following contents: the usefulness of emotions, the emotions and our body, the intensity of emotions, and the critical point. Moreover, the session trains on a breathing technique to favor relaxation.

Session 2: to reinforce the capability to identify negative thoughts and to understand the associations between thoughts, emotions, and behaviors. 

Session 3: to reinforce the usage of a problem-solving method presented in Reframe-IT (SMART Plan), which helps to assess the potential solutions and their consequences to improve outcomes. 

Session 4: To improve the capability of cognitive restructuring. Participants are encouraged to recognize negative automatic thoughts and related emotions to improve the identification and management of potential solutions. 

### 3.3. Criteria for Discontinuing or Modifying Allocated Interventions

Participants in the intervention group can leave the study at any time if they wish without any consequences; this means that their information and collected data will not be analyzed. On the other hand, the control group will keep its condition during the whole trial, but they also can leave the study at any time if they wish without any consequences.

### 3.4. Strategies to Improve Adherence to Interventions

Adherence to the intervention will be monitored by research assistants from the research team. School psychologists will fill out a small survey after each session informing the attendance of students and information about the quality of the delivery. Research assistants will be in close contact with school authorities and school psychologists, monitoring the progression of the program during study visits.

### 3.5. Provisions for Post-Trial Care

There is no potential harm or damage in this trial. The intervention group will receive an evidence-based intervention and the control group will receive the usual care in health centers.

### 3.6. Outcomes

Self-report questionnaires assessing primary and secondary outcomes will be administered at baseline (one week before the start of the intervention), at post-intervention, and follow-up (3, 6 and 12 months). Primary and secondary outcomes will be measured electronically using Qualtrics.

Participants will be asked sociodemographic information including, age, sex at birth, gender, race/ethnicity, education, family income, and psychological/psychiatric treatment.

#### 3.6.1. Primary Outcome

The primary outcome (suicide ideation) will be assessed by the Suicidal Ideation Questionnaire (SIQ) [63], a 15-item self-report measure designed to determine SI in adolescents. Scores are ranked 0–6, with higher scores indicating greater severity of suicidal ideation. The maximum score is 90 and a cut-off score of 31 indicates a clinically meaningful level of suicidal ideation. It has been validated with clinical and non-clinical populations and has shown high levels of internal consistency (α > 0.90) and test–retest reliability (α > 0.90) and high levels of construct and criterion validity [64].

#### 3.6.2. Secondary Outcomes and Questionnaires

Suicide attempt: two questions assessing whether participants had attempted suicide; if yes, how many attempts?

Anxiety as trait and as state: The State–Trait Anxiety Inventory (STAI-X) [65] is a self-report questionnaire with two scales of 20 items, each assessing state and trait levels of anxiety. Respondents indicate how they generally feel on a 4-point Likert scale from 0 (almost never) to 3 (almost always). The minimum score is 0 and the maximum score is 60 for both scales. Higher scores mean higher trait or state anxiety levels. It has been validated in the Chilean population aged 13–60 years old [66].

Hopelessness: The Beck Hopelessness Scale [67] is a self-report scale with 20 true or false items, nine of which are keyed ‘false’ and 11 are keyed ‘true’. For every statement, each response is assigned a score of 0 or 1 and the total hopelessness score is the sum of the scores on the individual items. The minimum score is 0 and the maximum score is 20. A higher score means higher hopelessness levels.

Social problem solving: The short form of the Social Problem-Solving Inventory Revised (SPSI-R Short Form) [68] is a 25-item self-report instrument measuring two adaptive problem-solving dimensions (positive problem orientation and rational problem solving) and three dysfunctional dimensions (negative problem orientation, impulsivity/carelessness style, and avoidance style). Each item is rated on a 5-point scale ranging from ‘not at all true of me’ (0) to ‘extremely true of me’ (4). The minimum score is 0 and the maximum score is 75. A higher score means a higher social problem-solving capability.

Cognitive-behavioral skills: The Cognitive-Behavioral Therapy Skills Questionnaire (CBTS) [69] is a 16-item scale with two scales measuring two skills: cognitive restructuring (CR) and behavioral activation (BA). Respondents rank each item on a 5-point Likert scale from 1 (I don’t do this) to 5 (I always do this). The minimum score is 15 and the maximum score is 80. A higher score means a higher presence of cognitive-behavioral skills. To assess general CBT skills, we have removed those items that refer exclusively to the practice of CBT sessions (e.g., keep track of the signs and symptoms of my condition); therefore, general CBT skills could be assessed before and after the intervention in both groups. 

Emotional regulation: The Emotion Regulation Questionnaire for Children and Adolescents (ERQ-CA) [70] is a 10-item self-report scale assessing two ER styles: Cognitive Reappraisal (CR, six items) Expressive Suppression (ES, four items). Items are rated on a 5-point Likert scale ranging from ‘strongly disagree’ (1) to ‘strongly agree’ (5). The minimum score is 10 and the maximum score is 50. A higher score means a higher presence of emotional regulation styles.

Adherence to TAU actions: A research assistant will register all the appointments and attendance of referred students to the primary care clinics. We will assess aspects related to the adherence to the actions included in the TAU.

### 3.7. Participant Timeline

Table 1 shows the participant timeline.

### 3.8. Sample Size

This sample size considers a loss at follow-up of 20% of the students. Considering a 20% of school rejection at the recruitment stage, we expect to recruit 12 schools. Studies of similar school-based interventions report effect sizes of at least 0.4 standard deviation using symptoms questionnaires as proposed here [71]. We used the results previously observed by Hetrick et al. [49] and the clustersampsi command in Stata Software to estimate the number of clusters in two arms, using the following command: clustersampsi, samplesize mu1(X1) mu2(X2) sd1(Y1) sd2(Y27) m(Z) rho(R). Where X1 = mean in arm 1; X2 = mean in arm 2; Y1 = standard deviation (SD) in arm 1; Y2 = SD in arm 2; Z = number of children per school on average (harmonic mean) (*n* = 14); and R = intracluster correlation. The two arms will be balanced for school size. 

### 3.9. Recruitment

The strategies for achieving adequate school enrolment to reach the target sample size will include contacting, and presenting the study to, municipality authorities, who are expected to help to establish contact with school authorities. Additionally, research assistants will contact directly and inform school authorities about the purpose, requirements, and duration of the study. Our research team already has good collaborative networks with municipalities and schools in the cities where the study will be conducted.

## 4. Assignment of Interventions: Allocation

Randomization will be performed once all schools are recruited. Schools will be randomly assigned to either group with a 1:1 allocation as per computer-generated randomization. In order to reduce the risk of allocation bias, the size of the schools and the vulnerability index will be taken into account to balance the groups. 

### 4.1. Sequence Generation

Randomization will be performed once all schools are recruited. Schools will be randomly assigned to either group with a 1:1 allocation as per computer-generated randomization.

### 4.2. Concealment Mechanism

After the randomization and allocation, schools will be informed by a research assistant of the group of belonging by email and confirmed by telephone. Each school will only receive their information of allocation. Additionally, this information will not be disclosed to the assessment research team (outcome evaluators) to keep the blind to the school allocation.

### 4.3. Allocation Implementation

An independent statistician will perform the randomization to assign to the study arms. This statistician will give this information to a research assistant. Later, the research assistant will inform the schools by email and confirm by telephone.

## 5. Assignment of Interventions: Blinding

### 5.1. Who Will Be Blinded?

This is a double-blinded trial, blinded to the outcome evaluators and to the data analyst. Outcome evaluators will not be informed about the group of belonging and will be instructed not to ask for the school condition to the students nor to the school authorities. A data analyst will work with the final dataset, where the group condition will be masked.

### 5.2. Procedure of Unblinding If Needed

The design is open-label with only outcome assessors and data analysts being blinded, so unblinding will not occur. Additionally, by the nature of the intervention, the participants are not blind to the group in which they belong. The evaluations will be carried out through self-report questionnaires sent in electronic format and their evaluation will be automatic without human action. In data analysis, the statistician will be blind to the intervention group of participants.

## 6. Data Collection and Management

### 6.1. Plans for Assessment and Collection of Outcomes

Self-report questionnaires assessing primary and secondary outcomes will be administered at baseline (one week before the start of the intervention), at post-intervention, and follow-up (3, 6, and 12 months). Primary and secondary outcomes will be measured electronically using Qualtrics. 

### 6.2. Plans to Promote Participant Retention and Complete Follow-Up

The schools, students, and their families will receive extensive information about the study setup and requirements during the recruitment. This information will include and stress the importance of completion of follow-up. From the start of the implementation of the assessments, students will be reminded of the value of their active participation during the whole project. Throughout the follow-up period, the researchers will check responses and if necessary contact schools and participants for completion of their follow-up. 

### 6.3. Data Management

After the participants have completed the online questionnaires, we will enter the data into a secure platform without identifying information (each participant will be assigned an encrypted ID number). Only the principal investigator, the research assistants in charge of data entry, and the statistician will have access to the database. All people with access to the dataset will need to sign a confidential agreement to assure the commitment to not revealing identifying information.

## 7. Statistical Methods

### 7.1. Statistical Methods for Primary and Secondary Outcomes

We will use descriptive statistics to assess balance across arms at baseline. The primary between-group analysis will be carried out on an intention-to-treat basis for 3-month SIQ scores. We will use linear mixed effects model analysis to compare the intervention and control groups regarding the change in outcome measures from baseline to post-intervention and the follow-up time point. The school is included as a random factor for each outcome measure to account for possible school effect. We will use Fisher’s exact test to compare the frequency of suicide attempt cases between groups. We also will conduct a Complier Average Causal Effect (CACE) analysis to assess the impact of the number of sessions on the primary outcome. All analyses will be performed using STATA 15.01. 

### 7.2. Statistical Methods for Additional Analyses

We will fit multilevel multiple mediation models in Mplus 6 [72], entering all hypothesized mediators simultaneously [73]. The conceptual model of mediation is depicted in Figure 1. 

We will calculate the intervention effect for each mediator, the effect of each mediator on the outcome (suicidal ideation), and the intervention effect that was mediated by each mediator. For each model, we will compute the total indirect effects of the intervention (i.e., the total effect of intervention mediated by the sum of the mediating factors) and the single indirect effect for each mediator as suggested by Preacher and Hayes [73]. Moreover, we will calculate the program’s direct effect (i.e., the effect not explained by the hypothesized mediators). The randomization occurred at the school level; therefore, we will enter school as the second level and individuals as the first level. To control for variability across schools, we will use the stratification option in Mplus. 

### 7.3. Interim Analyses

There will not be interim analyses because the data will be analyzed at the end of the trial.

### 7.4. Methods in Analysis to Handle Protocol Non-Adherence and Any Statistical Methods to Handle Missing Data

The primary outcome will be assessed using an intention-to-treat analysis. Missing data will be reduced to a minimum by using the appropriate measures; encouraging students to fill out the whole questionnaire, a research assistant will revise the questionnaire when students end the evaluation and ask to complete the instruments if some questions are unanswered. Multiple imputations will be used to handle missing data in the primary and secondary analyses.

### 7.5. Plans to Give Access to the Full Protocol, Participant Level-Data, and Statistical Code

The data set produced during the study will be available in an international database repository (UK Data Service).

## 8. Oversight and Monitoring

### 8.1. Composition of the Coordinating Center and Trial Steering Committee

Daily support for the trial will be provided by the principal investigator, who takes supervision of the trial. Additionally, there is a data manager who organizes data collection and assures data quality. The study coordinator helps in trial registration and will coordinate study visits and reports. Study research assistants will help to identify potential participating schools, collect informed consents, ensure follow-up according to protocol, and resolve doubts from school staff or participants. 

The study team will meet weekly during the whole duration of the study. There is no trial steering committee or stakeholder and public involvement group. The Ethical Scientific Committee of the Universidad de Talca will check the presence and completeness of the investigation.

### 8.2. Composition of the Data Monitoring Committee, Its Role, and Reporting Structure

A monitor from the Ethical Scientific Committee of the Universidad de Talca will check once a year the presence and completeness of the investigation. This committee is independent of the sponsor and has no competing interests; further details about its charter can be asked via email: cec@utalca.cl.

### 8.3. Adverse Event Reporting and Harms

The intervention does not infer harm among the participants. However, any situation that compromises the physical and psychological integrity of participants occurring during the intervention process will be reported by the psychologist working with the students by filling out a predesign form. This professional will activate the protocol for managing suicidal behavior in schools already in place in all educational institutions in Chile (MINSAL, 2019). This protocol includes contacting school authorities, main caregivers, and local health providers in order to obtain a quick response from health services. It also includes a follow-up and case management. In parallel, the psychologist will inform the principal investigator and the project coordinator, who will support the whole process. To reduce the risk of low implementation of this protocol, we will train the school personnel before the intervention. 

### 8.4. Frequency and Plans for Auditing Trial Conduct

A monitor from the Ethical Scientific Committee of the Universidad de los Andes will check once a year the presence and completeness of the investigation files such as informed consents, inclusion and exclusion criteria, and data collection and storage.

### 8.5. Plans for Communicating Important Protocol Amendments to Relevant Parties (e.g., Trial Participants, Ethical Committees)

All substantial amendments will be notified to the Ethics Committee of the Universidad de Talca. In case amendments concern or affect participants in any way, they will be informed about the changes. If needed, additional consent will be requested and registered. Additionally, online trial registries will be updated accordingly.

### 8.6. Dissemination Plans

The results of this research will be disclosed completely in international peer-reviewed journals. Both positive and negative results will be reported. An executive summary of the results will be given to school authorities.

## 9. Discussion

This study is the first in Latin America to test the effectiveness of a blended school-based intervention to reduce suicidal ideation among adolescents in a cluster randomized controlled trial (cRCT). The whole intervention proposed in this protocol was based on the promissory results of the internet-based component (Reframe-It) and the evidence showing the need to include face-to-face therapeutic sessions. If the effects of the intervention are positive, wide implementation in Chile and other Latino American countries could be possible in the future.

This project faces several challenges related to feasibility and the implementation to be addressed in order to help the scaling-up process in the future. For instance, this project requires a large sample size, which implies on the one hand a high demand in time and effort for the research team and on the other hand a high degree of motivation and involvement from the schools. Regarding the latter, sometimes it is not easy for the schools to accept the degree of commitment and workload needed to conduct this kind of project. To minimize these risks, we will prepare the recruitment and the training for the schools carefully and inform them in good time of all the steps and milestones of the project.

## 10. Conclusions

We expect to test the effectiveness of a blended CBT intervention to reduce suicidal ideation in adolescents as the primary outcome. Moreover, we expect to provide evidence on the potential mediation role of hopelessness, emotional regulation, and problem-solving skills on the effect of the intervention to reduce suicidal ideation.

## Figures and Tables

**Figure 1 ijerph-19-03947-f001:**
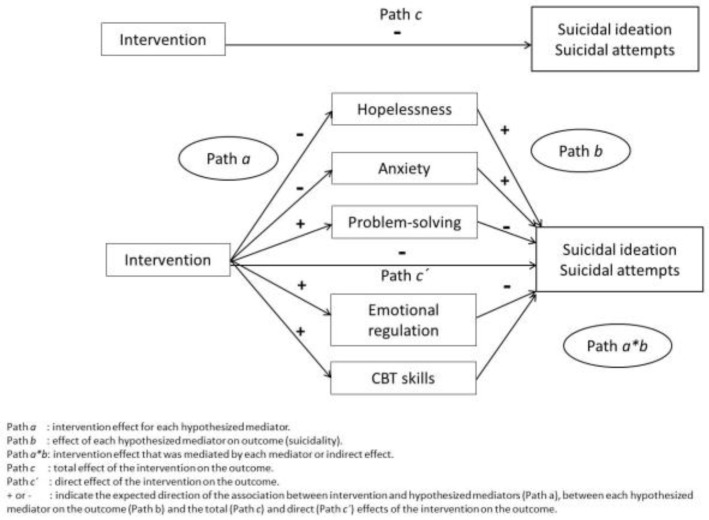
Conceptual model of mediation.

**Table 1 ijerph-19-03947-t001:** Standard Protocol Items: recommendations for International Trials (SPIRIT) diagram.

	STUDY PERIOD																			
TIMEPOINT	2022						2023											2024		
	Enrolment				Allocation		Post-Allocation											Follow-Up		Close-Out
	Aug.	Sep.	Oct.	Nov.	Dec.	Jan.	Feb.	Mar.	Apr.	May.	Jun.	Jul.	Aug.	Sep.	Oct.	Nov.	Dec.	Mar.	Sep.	Oct.
**ENROLMENT**																				
Contact with schools	X	X	X	X																
Eligibility screen			X	X	X															
Allocation					X															
Schools are informed about group belonging					X	X														
Informed consent								X												
**INTERVENTIONS**																				
Training								X												
Blended intervention											X	X	X	X						
**ASSESSMENTS**																				
**Baseline variables** (descriptive features, suicide ideation, suicide attempts, anxiety, hopelessness, social problem and cognitive-behavioral skills, emotional regulation)										X										
**Outcome variables** (suicide ideation, suicide attempts, anxiety, hopelessness, social problem and cognitive-behavioral skills, emotional regulation)														X			X	X	X	X
**DISSE** **MINATION**																	X	X	X	X

X = milestone.

## Data Availability

Not applicable.

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
