# Peer review of "Testing the Effectiveness of a Blended Intervention to Reduce Suicidal Ideation among School Adolescents in Chile: A Protocol for a Cluster Randomized Controlled Trial"

_ijerph, 2022, doi:10.3390/ijerph19073947_

Round 1
Reviewer 1 Report
The manuscript at present is a draft of a research design, it is not a research, nor is it a protocol. Its nature is still ambiguous. A substantial and clear description of the intervention that is to be implemented and then verified through this research is missing.
If you intend to write a protocol it is important to declare the hypotheses within which the intervention was built and to use them in the verification of the intervention that you want do in current paper. The manuscript requires a structural change, particularly of the first part. Starting from a novel strong introduction, a greater reasoned conclusion must be built.
Author Response
Response to Reviewer 1 Comments
- The manuscript at present is a draft of a research design, it is not a research, nor is it a protocol. Its nature is still ambiguous.
Response: We rearranged the manuscript following the general structure of the Standard Protocol Items: Recommendations for Interventional Trials (SPIRIT; https://www.spirit-statement.org/spirit-statement/)
- A substantial and clear description of the intervention that is to be implemented and then verified through this research is missing.
Response: We added some specific information about the intervention in the Introduction section (page 3). Additionally, we included a better description of the topics covered by each module of the program, in the Intervention section (page 6).
- If you intend to write a protocol it is important to declare the hypotheses within which the intervention was built and to use them in the verification of the intervention that you want do in current paper.
Response: We added the following paragraph describing the hypotheses in the Introduction section: “We hypothesize that the intervention would reduce suicidal ideation among the treatment group compared to the control group at post-intervention. Specifically, we hypothesize that the intervention will lead to the following changes among the treatment group compared to the control group at post-intervention and follow-up: reduced feelings of depression, anxiety, and hopelessness, increased emotional regulation, and problem-solving skills. Finally, we hypothesize that the effect of the integrated blended intervention on suicidal ideation will be mediated by hopelessness, emotional regulation, and problem-solving skills, and partially mediated by CBT skills acquired from the intervention. The results of this study will provide relevant insights to reinforce direct and indicated interventions in educational settings”.
- The manuscript requires a structural change, particularly of the first part. Starting from a novel strong introduction, a greater reasoned conclusion must be built.
Response: We have rearranged the introduction including the following information on the rationale of the study: “The review by Büscher et al. (2020) showed that the only direct intervention targeting suicidal ideation in school students with suicidal ideation is the Reframe-IT, developed by Robinson et al. (2014). This intervention was built following the main principles of CBT, considering the evidence that cognitions are central in the development and maintenance of mental health difficulties and that emotional and behavioral problems can be resolved by modifying dysfunctional thoughts (Kazantzis et al., 2018). The intervention is organized in eight modules to promote the following skills: problem identification, emotional recognition and distress tolerance, behavioral activation, cognitive restructuring, distress tolerance, and problem-solving skills” (page 3).
We included the objectives in the Introduction section: “The main objective of this study is to develop a blended intervention to reduce suicidal ideation and to test its effectiveness among adolescents in secondary schools in Chile at post-intervention and 3-month, 6-month, and 12-month follow-up. Additionally, the study will identify and compare changes in depressive and anxiety symptoms, hopelessness, emotional regulation, and problem-solving skills among adolescents in the intervention group schools and the control group. Finally, we will explore the influence of potential mediators on the observed effects of the integrated blended intervention” (page 4).
Moreover, as we previously stated, we added the trial hypotheses.
Finally, we have included the trial design in the Introduction: This is a protocol for a single-blind, two-armed randomized controlled trial evaluating changes in primary and secondary outcomes post-intervention and follow-up. The two arms will be (a) the intervention, which comprises eight internet-based modules of CBT delivered over 10-week using the program Reframe-IT, and four 45-min face-to-face psychotherapeutic CBT sessions; and (b) Control group (Treatment-As-Usual) (see Table 1).

Reviewer 2 Report
All comments addressed.
Author Response
Dear reviewer,
Thank you for your comments
This manuscript is a resubmission of an earlier submission. The following is a list of the peer review reports and author responses from that submission.
Round 1
Reviewer 1 Report
Thank you for the opportunity to review this protocol to test the effectiveness of a blended intervention on suicide ideation in adolescents using a cRCT, and in addition the potential mediators.
Overall the manuscript of the protocol is robust and comprehensive.
The following are methodological points to address:
- page 3, line 96 specify if this effect was reduced when measured after the 8 weeks or also in follow-up measurements
- 2.1.4 line 218, 'inclusion' should be replaced with 'exclusion' and add to the inclusion criteria: inclusion for C-SSRS greater or equal to 3 but not higher than 30?? (as cut-off score of 31 indicates a clinically meaningful level of suicidal ideation and these participants will not be included, but referred immediately for help. Is that correct?
- 2.2 Intervention: participants will follow the protocol for suicide ideation from the Ministry / this wording is misleading (as if we want students to follow a protocol on how to do suicide ideation) so rewrite and start with the intervention text to match the heading and the TAU text moves to 2.2
- Conclusion: also add the test of what mediators are at play in the effect
- General: replace 'developing countries' with 'low-and middle-income or low-income'
In addition, a professional, scientific, native English proofreader is required to improve the writing, e.g.:
-Abstract line 18, delete 'Accordingly, by a single blind' as it is difficult to follow and replace with for example: Thus, we will tests ...using a single blind.....
-Abstract lines 25 to 27 repeat the same terms so make more concise and add a period at the end of the abstract.
-Page 2, line 41 to 45 is too long so hard for reader to follow, make into two sentences.
-Section 2.1.4 Line 196, this is not a complete sentence
-Line 225 General should be lower case 'g' and also for section 2.2.3 line 182
-Periods at end of sentences missing throughout.
Author Response
Response to Reviewer 1 Comments
- Page 3, line 96 specify if this effect was reduced when measured after the 8 weeks or also in follow-up measurements.
Response: We included the required specification: After 8 weeks (post-intervention).
- 1.4 line 218, 'inclusion' should be replaced with 'exclusion' and add to the inclusion criteria: inclusion for C-SSRS greater or equal to 3 but not higher than 30?? (as cut-off score of 31 indicates a clinically meaningful level of suicidal ideation and these participants will not be included, but referred immediately for help. Is that correct?
Response: We replaced 'inclusion' as the reviewer suggested. Concerning the inclusion of additional criterium, we think that it is not necessary. We used the first seven items of the Columbia Suicide Severity Rating Scale (C-SSRS) for screening and eligibility purposes. Then, participants with scores greater or equal 3 will be invited to participate into the study. Thereafter, the Suicidal Ideation Questionnaire (SIQ) will be used to measure the primary outcome (suicidal ideation) of those who accept to participate in the study. In both cases, students with clinically meaningful level of suicidal ideation will be referred to health care centers, following the protocol defined by the Chilean Ministry of Health.
2.2 Intervention: participants will follow the protocol for suicide ideation from the Ministry / this wording is misleading (as if we want students to follow a protocol on how to do suicide ideation) so rewrite and start with the intervention text to match the heading and the TAU text moves to 2.2
Response: We edited the text, and we started the paragraph with the text with intervention description. We additionally modified and moved the text about TAU to 2.1.3: “All participants in the intervention group will be sent to primary care centers, where a protocol for managing suicidal ideation defined by Chilean Ministry of Health will be applied (MINSAL, 2019). The whole procedure will be described as Treatment-As-Usual (TAU). TAU consists of identification and assisted referral to a primary care clinic where trained psychologists will assess the symptomatology and propose a course of actions, from initiation of psychotherapy to referral to a General practitioner to initiate medication if needed. Psychotherapy in primary care clinics consists of a range of 4-8 sessions twice a month. Most of the content of the psychotherapy is based on Cognitive-Behavioral Model. When needed, general practitioners may suggest using medication, which generally are SSRI antidepressants (e.g., Fluoxetine, Sertraline). A medical check-up is conducted every month or every two months” (page 6).
- Conclusion: also add the test of what mediators are at play in the effect
Response: We added the following sentence: “Moreover, we expect to provide evidence on the potential mediation role of hopelessness, emotional regulation, and problem solving skills on the effect of the intervention to reduce suicidal ideation.”
- General: replace 'developing countries' with 'low-and middle-income or low-income':
Response: 'developing countries' was replaced throughout the manuscript with the expression 'low-and middle-income'. See page 2.
- In addition, a professional, scientific, native English proofreader is required to improve the writing, e.g.:
Response: The whole manuscript has been proof-read by native English professional.
6.1 Abstract line 18, delete 'Accordingly, by a single blind' as it is difficult to follow and replace with for example: Thus, we will tests ...using a single blind…
And,
6.2 Abstract lines 25 to 27 repeat the same terms so make more concise and add a period at the end of the abstract.
Response to 6.1 and 6.2: We have modified the abstract as follow: “Thus, we will test the effectiveness of a blended intervention to reduce suicidal ideation (primary outcome) in school settings using a single-blind two-armed cluster randomized controlled trial (cRCT).” The internet-based component corresponds to the Reframe-IT, a program encompassing eight online sessions based on cognitive-behavioral therapy (CBT) principles. The face-to-face intervention will be delivered through four CBT sessions. Additionally, we will assess the effect of the intervention on the following secondary outcomes: suicidal attempts, depressive symptoms, hopelessness, emotional regulation and problem-solving skills. Primary and secondary outcomes will be assessed at post-intervention, 3-month, 6-month, and 12-month follow-up. Finally, we will explore the mediation role of cognitive, emotional and behavioral correlates of suicide on the effect of the intervention. Results will inform whether the intervention can reduce suicide among school adolescents and be implemented on a large scale in Chile.”
6.3 Page 2, line 41 to 45 is too long so hard for reader to follow, make into two sentences.
Response: We have edited and changed the sentence as follow: “Most adolescents receive a formal education (Macleod et al., 2015). Moreover, school-based interventions can have a positive impact on SRB and, overall, do not appear to cause harm (Robinson et al., 2018). Thus, schools are currently considered as accepted context to implement suicide-prevention programs (Breux et al., 2019; Singer et al., 2018).
6.4 Section 2.1.4 Line 196, this is not a complete sentence
Response: We modified the sentence: “Participants will be adolescents attending schools with secondary education (Grades 9-11), mixed-sex, located in three different cities in the South of Chile, with at least two classes per year”.
6.5 Line 225 General should be lower case 'g' and also for section 2.2.3 line 182
Response: It has been modified throughout the manuscript.
6.6 Periods at end of sentences missing throughout.
Response: Periods were added.

Reviewer 2 Report
The authors presented a clinical trial protocol that is aimed at investigating the efficiency of blended interventions to reduce suicide ideation in adolescents. While the protocol presents a strong methodology for cluster RCT improvements are needed
- This is a study protocol (not a regular article with results), which must be stated in the title and clearly in the introduction.
- Has the recruitment started? if yes indicate that in the manuscript.
- A diagram to show the methodology and assessment timing will be good.
- Page 5 L207 and 214. Both are inclusion criteria and how are they different?
- State the exclusion criteria
- How would you handle adverse events? a plan needs to be indicated since the study is about suicide ideation what happens if any of the study subject attempt suicide during the study period? would you break confidentiality in such cases? A detailed plan is needed and reported.
- Information on data management is needed since it is stated that the statistician is blinded.
- how would you account for allocation and performance bias?
- Discussion is too short with no relevant information. What is the strength and limitations of this protocol? what are the challenges you are anticipating.
- Are all outcomes measured electronically? if yes state that. if not how would you handle detection bias?
Author Response
Response to Reviewer 2 Comments
- This is a study protocol (not a regular article with results), which must be stated in the title and clearly in the introduction.
Response: We modified the title of the manuscript as follow: “Testing the effectiveness of a blended intervention to reduce suicidal ideation among school adolescents in Chile: A protocol for a cluster randomized controlled trial”. Moreover, we added the following sentence at the end of the Introduction section: “In summary, in the context of the recognized need to improve the effectiveness of interventions targeting suicide ideation in adolescents, we developed a study protocol of a blended intervention combining an internet-based intervention (Reframe-IT) with 4 face-to-face sessions based on CBT principles. The results of this study will provide relevant insights to reinforce direct and indicated interventions in educational settings”. Finally, we slightly modified the Design section, to clarify that this is a study protocol: “This is a protocol for a single-blind, two-armed randomized controlled trial evaluating changes in primary and secondary outcomes post-intervention and follow-up”.
- Has the recruitment started? if yes indicate that in the manuscript.
Response: The recruitment process has not started. It will start on August, 2022.
- A diagram to show the methodology and assessment timing will be good.
Response: The diagram showing the methodology and assessment timing was included into the manuscript (page 4).
- Page 5 L207 and 214. Both are inclusion criteria and how are they different?
Response: This has been corrected, and now, it is possible to differentiate inclusion from exclusion criteria.
- State the exclusion criteria
Response: This has been corrected, and now, it is possible to differentiate inclusion from exclusion criteria.
- How would you handle adverse events? a plan needs to be indicated since the study is about suicide ideation what happens if any of the study subject attempt suicide during the study period? would you break confidentiality in such cases? A detailed plan is needed and reported.
Response: We added a specific section about the management of potential adverse events (page 9):
“The intervention does not infer harm among the participants. However, any situation that compromises the physical and psychological integrity of participants occurring during the intervention process will be reported by the psychologist working with the students and filling out a pre-design form. This professional will activate the protocol for managing suicidal behavior in schools already in place in all educational institution in Chile (MINSAL, 2019). This protocol includes contacting school authorities, main care-givers, and local health providers in order to obtain a quick response from health services. It also includes a follow-up and case-management. In parallel, the psychologist will inform to the Principal Investigator and Project Coordinator, who will support the whole process. To diminish the risk of low implementation of this protocol, we will train the school personnel before the intervention.”
- Information on data management is needed since it is stated that the statistician is blinded.
Response: We added a new section “Data Management”, which states the following: “After the participants have completed the online questionnaires, the data will be entered into a secure platform without identifying information (each participant will be assigned an encrypted ID number). Only the Principal Investigator, research assistants in charge of data entry, and the statistician will have access to the database. All people with access to the dataset will need to sign a Confidential Agreement to assure its commitment to not revealing identifying information” (See page 8).
- How would you account for allocation and performance bias?
Response:
Regarding allocation bias, we have added the following text in page 5: “Randomization will be performed once all schools are recruited. Schools will be randomly assigned to either group with a 1:1 allocation as per computer-generated randomization. In order to reduce the risk of allocation bias, size of the schools and the vulnerability index will be taken into account to balance the groups. An independent statistician will perform the randomization.”
Regarding performance bias, we have added the following text in page 7: “In order to reduce the risk of performance bias, all adolescents who need to be cared in a public health center will received the same treatment. The public health centers have similar structure and resources, they have standardized protocols for managing mental disorders and suicide risk and professionals have with similar background and skills. To test potential biases, we will register and quantify all different interventions delivered by health centers.”
- Discussion is too short with no relevant information. What is the strength and limitations of this protocol? what are the challenges you are anticipating.
Response: We edited the Discussion section including the aspects requested by the reviewer: “This study is the first in Latin America to test the effectiveness of a blended school-based intervention to reduce suicidal ideation among adolescents in a cluster randomized controlled trial (cRCT). The whole intervention proposed in this protocol was based on the promissory results of the internet-based component (Reframe-It) and the evidence showing the need to include face-to-face therapeutic sessions. If the effects of the intervention are positive, wide implementation in Chile and other Latino American countries could be possible in the future.
This project faces several challenges related to feasibility and the implementation to be addressed in order to help the scaling-up process in the future. For instance, this project requires a large sample size, which implies, on the one hand a high demand in time and effort for the research team, and on the other hand, a high degree of motivation and involvement from the schools. Regarding the latter, sometimes it is no easy for the schools to accept the degree of commitment and workload needed to conduct this kind of projects. To minimize these risks, we will prepare the recruitment and the training for the schools carefully and inform them in good time all the steps and milestones of the project.”
- Are all outcomes measured electronically? if yes state that. if not how would you handle
Response: All outcomes will be measured electronically. We included the following sentence: “Primary and secondary outcomes will be measured electronically, using Qualtrics” (page 7).

Reviewer 3 Report
The article presents research that will have to be done in the future. There is no data collected, nor has an analysis of any kind been carried out. Furthermore, the paper does not even present itself as a review of the scientific literature. For this reason the paper cannot be accepted given the ambiguity of the paper.
I recommend writing only a review of the scientific literature at this stage and submitting the research when it is will be completed.
Author Response
The article presents research that will have to be done in the future. There is no data collected, nor has an analysis of any kind been carried out. Furthermore, the paper does not even present itself as a review of the scientific literature. For this reason the paper cannot be accepted given the ambiguity of the paper.
I recommend writing only a review of the scientific literature at this stage and submitting the research when it is will be completed.
Response: We modified the title of the manuscript as follow: “Testing the effectiveness of a blended intervention to reduce suicidal ideation among school adolescents in Chile: A protocol for a cluster randomized controlled trial”. Moreover, we added the following sentence at the end of the Introduction section: “In summary, in the context of the recognized need to improve the effectiveness of interventions targeting suicide ideation in adolescents, we developed a study protocol of a blended intervention combining an internet-based intervention (Reframe-It) with 4 face-to-face sessions based on CBT principles. The results of this study will provide relevant insights to reinforce direct and indicated interventions in educational settings.” Finally, we slightly modified the Design section, to clarify that this is a study protocol: “This is a protocol for a single-blind, two-armed randomized controlled trial evaluating changes in primary and secondary outcomes post-intervention and follow-up”.
